# Antenatal, intrapartum and infant azithromycin to prevent stillbirths and infant deaths: study protocol for SANTE, a 2×2 factorial randomised controlled trial in Mali

Amanda J Driscoll [ID],[1] Fadima Cheick Haidara,[2] Milagritos D Tapia,[1] Emily L Deichsel,[1] Ousmane S Samake,[2] Tiecoura Bocoum,[2] Jason A Bailey,[1] Meagan C Fitzpatrick,[1] Robert L Goldenberg,[3] Mamoudou Kodio,[2] Lawrence H Moulton,[4] Dilruba Nasrin,[1] Uma Onwuchekwa,[2] Allison M Shaffer,[1] Samba O Sow,[2] Karen L Kotloff[1]

AJD and FCH contributed equally.
SOS and KLK contributed equally.

► http://dx.doi.org/10.1136/bmjopen-2022-068487.R1

For numbered affiliations see end of article.

**Correspondence to**
Dr Amanda J Driscoll;
adriscoll@som.umaryland.edu

## ABSTRACT

**Introduction** In high mortality settings, prophylactic azithromycin has been shown to improve birth weight and gestational age at birth when administered antenatally, to reduce the incidence of neonatal infections when administered intrapartum, and to improve survival when administered in infancy. Questions remain regarding whether azithromycin can prevent stillbirths, and regarding the optimal strategy for the delivery of azithromycin to pregnant women and their infants.

**Methods and analysis** Sauver avec l'Azithromycine en Traitant les Femmes Enceintes et les Enfants (SANTE) is a 2×2 factorial, individually randomised, placebo-controlled, double-masked trial in rural Mali. The primary aims are: (1A) to assess the efficacy of antenatal and intrapartum azithromycin on a composite outcome of stillbirths and infant mortality through 6–12 months and (1B) to assess the efficacy of azithromycin administered concurrently with the first and third doses of pentavalent vaccines (Penta-1/3) on infant mortality through 6–12 months. Pregnant participants (n=49 600) and their infants are randomised 1:1:1:1 to one of four treatment arms: (1) mother and infant receive azithromycin, (2) mother and infant receive placebo, (3) mother receives azithromycin and infant receives placebo or (4) mother receives placebo and infant receives azithromycin. Pregnant participants receive three single 2 g doses: two antepartum and one intrapartum. Infants receive a single 20 mg/kg dose at the Penta-1 and 3 visits. An additional cohort of 12 000 infants is recruited at the Penta-1 visit and randomised 1:1 to receive azithromycin or placebo at the same time points. The SANTE trial will inform guidelines and policies regarding the administration of antenatal and infant azithromycin using routine healthcare delivery platforms.

**Ethics and dissemination** This trial was approved by the Institutional Review Board at the University of Maryland School of Medicine (Protocol #HP-00084242) and the Faculté de Médecine et d'Odonto-Stomatologie in Mali. The findings of this trial will be published in open access peer-reviewed journals.

**Trial registration number** NCT03909737.

## STRENGTHS AND LIMITATIONS OF THIS STUDY

⇒ The trial is conducted in a low-resource setting with high rates of stillbirth and infant mortality.
⇒ The intervention is delivered through the public healthcare system, relying on the existing infrastructure for routine primary care.
⇒ The trial assesses multiple time points for the administration of azithromycin to prevent infant mortality (antenatal, intrapartum and infant).
⇒ The trial includes ancillary studies to assess antimicrobial resistance and to investigate potential mechanisms of action of azithromycin in subsets of participants.
⇒ The trial is unable to reliably determine individual causes of stillbirth and infant death.

## INTRODUCTION

Despite significant improvements in child survival over the past two decades, global rates of stillbirth and infant mortality remain unacceptably high. Of the 5.2 million under-5 deaths in 2019, 3.9 million (75%) occurred within the first 12 months of life and 2.4 million (45%) within the first month.[1] More than half of global under-5 deaths and 42% of the estimated 2 million global stillbirths occur in sub-Saharan Africa, where approximately 60% of child mortality is estimated to be due to preventable infectious causes.[2 3] Within the Child Health and Mortality Prevention Surveillance (CHAMPS) network, post-mortem sampling identified at least one infectious condition in the causal chain leading to death in 53% of neonatal deaths and 17% of stillbirths.[4] However, the causes of most stillbirths in low-income and middle-income

countries are unexplained, owing to insufficient data and the failure to adopt a standardised classification system for perinatal cause of death.[5–7] While malaria and syphilis are the most commonly recognised infectious causes of stillbirth, postmortem sampling in CHAMPS identified *Escherichia coli/Shigella*, *Streptococcus agalactia* and *Enterococcus faecalis* as other potentially significant infectious causes.[4 8]

An effective new strategy to prevent stillbirths and infant deaths that could be incorporated into routine primary healthcare services would be a major achievement in global health. One promising intervention to improve child survival is the prophylactic administration of the antibiotic azithromycin, a second-generation broad-spectrum macrolide that is used in mass drug administration (MDA) campaigns to control trachoma.[9] Current WHO guidelines recommend consideration of biannual MDA-azithromycin for infants aged 1–11 months to improve survival in high mortality settings.[10] These guidelines are based on data from three randomised controlled trials (RCTs), two of which found a statistically significant reduction in all-cause mortality among children randomised to receive azithromycin.[11–13] There are insufficient data to support global policy recommendations for prophylactic azithromycin in pregnancy, although it has also been shown to reduce the incidence of preterm birth and low birth weight when administered antenatally, and to prevent neonatal infections when administered intrapartum.[14 15] Further evidence is needed to understand (1) whether antenatal and intrapartum azithromycin can prevent stillbirths and early infant deaths, (2) the optimal method and timing of azithromycin administration in infancy to prevent mortality and (3) whether the survival benefit observed in two randomised MDA-azithromycin trials can be replicated in other settings.

Here, we describe the protocol for an individually randomised, placebo-controlled, 2×2 factorial trial to determine (1) the effect of antenatal and intrapartum azithromycin on stillbirths and infant mortality and (2) the effect of azithromycin on infant mortality when delivered with the first and third doses of diphtheria, tetanus, pertussis, hepatitis and Hib-containing (ie, pentavalent) vaccine. The trial will take place in Mali, a low-income West African country with high stillbirth, neonatal and infant mortality.

## METHODS AND ANALYSIS
### Study design, objectives and hypotheses
Sauver avec l'Azithromycine en Traitant les Femmes Enceintes et les Enfants (SANTE) is an individually randomised, placebo-controlled, double-masked trial. Using a 2×2 factorial design with four treatment arms, SANTE will simultaneously assess coprimary aim 1A: whether azithromycin administered at routine antenatal care (ANC) visits (second and/or third trimester) and at delivery reduces a composite outcome of stillbirth and infant mortality, and coprimary aim 1B: whether

azithromycin administered to infants concurrently with the first and third doses of pentavalent vaccine (Penta-1 and Penta-3) reduces mortality from the time of the first dose through 6–12 months of life. We will enrol 49 600 pregnant participants who will be randomised to receive 2 g azithromycin or placebo as a single oral dose twice during pregnancy and at delivery, for up to three total doses. Their infants will be randomised to receive 20 mg/kg azithromycin or placebo as a single oral dose at their routine Penta-1 and Penta-3 vaccination visits (figure 1). According to the national routine immunisation schedule, these vaccination visits occur at approximately 6 and 14 weeks of age, respectively. Infants are followed through at least 6 months and up to 12 months of age. To increase statistical power for aim 1B, a separate cohort of 12 000 infants, the infant-only cohort, is recruited at the Penta-1 visit. Infants enrolled in the infant-only cohort are randomised, dosed and followed in the same manner as those born to pregnant participants. Participant enrolment commenced in September 2020 and is expected to be complete by April 2023.

We hypothesise that the composite outcome of stillbirths and infant deaths will be lower in infants whose mothers were randomised to receive azithromycin in pregnancy compared with those randomised to receive placebo, and that mortality will be lower in infants who were randomised to receive azithromycin at first and third pentavalent vaccination visits compared with those randomised to receive placebo.

The trial has been registered at ClinicalTrials.gov (NCT03909737). The WHO trial registration data set can be found in online supplemental appendix table 1. Study recruitment began in September 2020 and is anticipated to end by April 2023. All follow-up will be completed within 12 months of enrolling the last pregnant participant.

### Study setting
SANTE is conducted by investigators at the Centre pour le Développement des Vaccins, Mali (CVD-Mali), an ongoing collaboration between the University of Maryland School of Medicine and the Malian Ministry of Health. For the past decade, much of northern and central Mali has been besieged by political instability and violence. The southwestern region of Sikasso, recently divided into the independent regions of Sikasso and Koutiala, had a reported infant mortality rate of 67 per 1000 live births in 2018.[16] It was selected as the study setting due to its high under-5 mortality, moderate utilisation of ANC, moderate routine immunisation coverage, and relative security (online supplemental appendix table 2). Malaria is endemic in southwestern Mali, and standard of care for pregnant women is at least three doses of sulfadoxine-pyrimethamine (SP) intermittent preventive treatment (IPTp). Children aged 3–59 months receive a combination of monthly SP plus amodiaquine as seasonal malaria chemoprophylaxis (SMC) for 4 months during peak malaria transmission season (July–October). Each

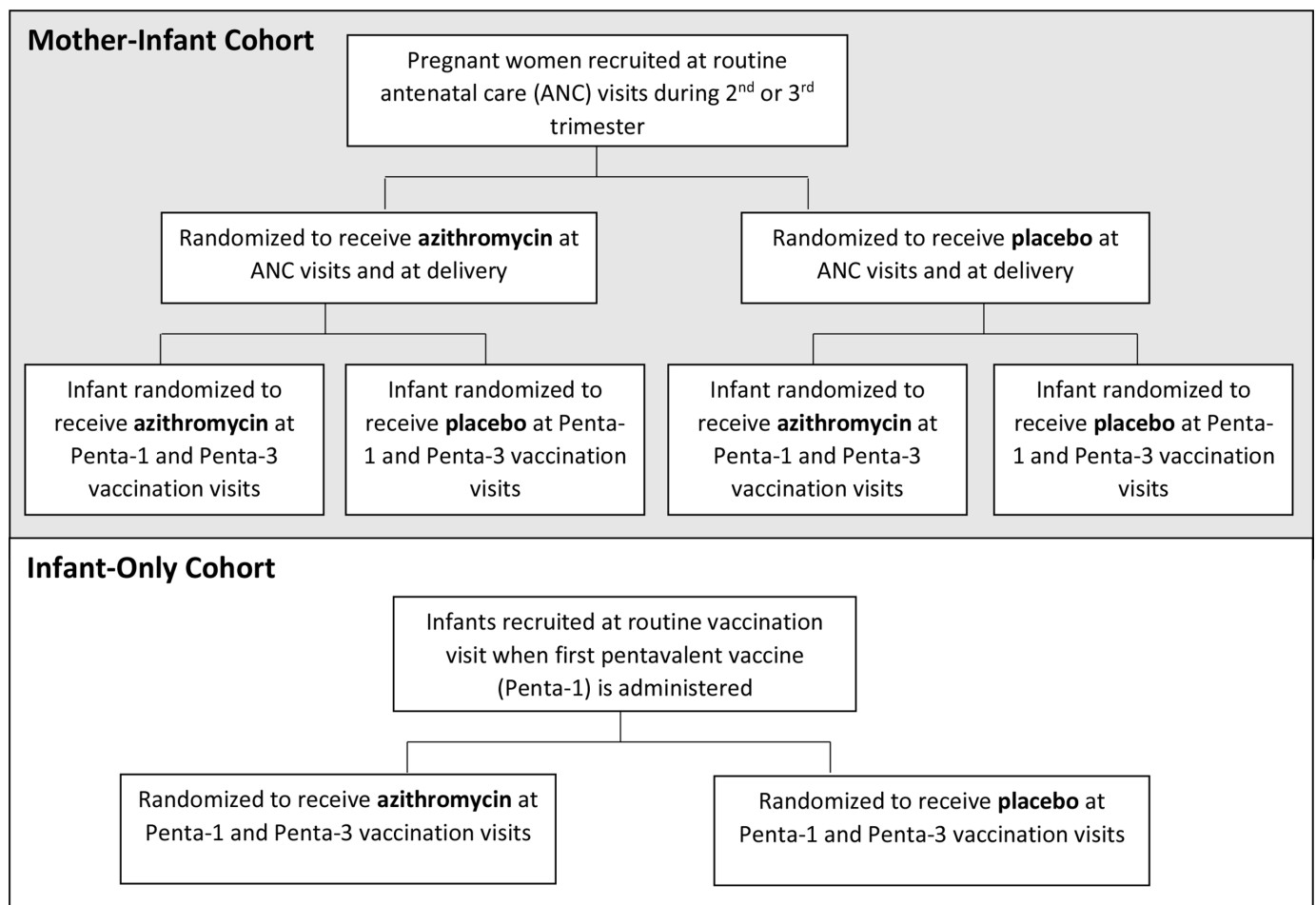

**Figure 1** SANTE study design. Flow diagram to illustrate randomisation and treatment groups for the 2x2 factorial mother–infant cohort and the infant-only cohort. 'Penta 1' and 'Penta 3' correspond to the first and third doses of pentavalent vaccine (ie, diphtheria, tetanus, pertussis, hepatitis and Hib-containing vaccine). SANTE, Sauver avec l'Azithromycine en Traitant les Femmes Enceintes et les Enfants.

monthly SMC dose is composed of one directly observed dose and two additional doses to be administered at home by the caregiver. In Sikasso, IPTp coverage with three or more doses is 26.8%, and 54% of age-eligible children receive one or more doses of SMC each year.[16] Within the Koutiala and Sikasso regions, the districts of Koutiala, Kignan and Niena were selected for study operations. These districts represent areas with a high burden of paediatric illness and have populations of sufficient size to meet the study's enrolment requirement within the target time frame.

### Recruitment

Mali's decentralised health system operates at national, regional, district (cercle), health area (aire de santé) and community levels. Each district is served by a referral centre (Centre de Santé de Référence, CSREF) and each aire de santé is served by one or more community health centres (Centres de Santé Communautaire, CSCOMs). SANTE study teams are embedded at all operational and accessible CSREFs and CSCOMs in the Koutiala, Kignan and Niena districts. Many rural CSCOMs conduct regular community outreach operations (stratégie avancée),

sending mobile units to remote health outposts to provide ANC and immunisation services. Study participants are recruited at antenatal or immunisation visits at participating CSREFs and CSCOMs, or from remote health posts as part of the stratégie avancée.

### Inclusion and exclusion criteria

Pregnant individuals seeking ANC are eligible to be enrolled if they have reached at least 14 weeks gestation and are not in active labour. If ultrasound dating is unavailable, participants are deemed eligible if the fundus can be palpated at two or more finger breadths above the pubic symphysis, a proxy measurement for a gestational age of at least 14 weeks.[17] Infants in the infant-only cohort are eligible to be enrolled if they are at least 6 weeks and fewer than 12 months of age and are presenting to a study facility to receive Penta-1 vaccine. Additional enrolment criteria are provided in table 1.

### Randomisation and masking

Treatment allocation is stratified by enrolment facility and randomised in permuted blocks of varying sizes (online supplemental appendix). In the mother-infant

**Table 1** Inclusion and exclusion criteria

| Enrolment criteria: Pregnant women and unborn infants enrolled in mother–infant cohort | |
|---|---|
| **Inclusion criteria** | **Exclusion criteria** |
| ▶ Participant is estimated to be ≥14 weeks gestation according to the study algorithm and is presenting at a study facility | ▶ Participant is in active labour, as defined by study standard operating procedures |
| ▶ Participant understands and can comply with planned study procedures, as determined by the study staff | ▶ Participant has a known allergy to macrolide antibiotics |
| ▶ Participant or legal authority has provided informed consent prior to initiation of any study procedures | ▶ Participant has a condition that, in the opinion of the investigator, might compromise his or her well-being, or affect the ability of the participant to comply with study procedures |
| ▶ Participant intends to reside in the study area until her newborn infant is at least 6 months old | ▶ Participant is currently being treated with a macrolide antibiotic for a medical condition (this can be a temporary exclusion if the drug is later discontinued) |
| **Enrolment criteria: Infants enrolled in the infant-only cohort** | |
| **Inclusion criteria** | **Exclusion criteria** |
| ▶ Participant is presenting for Penta-1 vaccination at a study facility | ▶ Participant has a known allergy to macrolide antibiotics |
| ▶ Participant is at least 6 weeks of age, and fewer than 12 months of age | ▶ Participant has a condition that, in the opinion of the investigator, might compromise his or her well-being, or affect the ability of the participant to comply with study procedures |
| ▶ Participant's parent/guardian understands and can comply with the study procedures, as determined by the study staff | ▶ Participant is currently being treated with a macrolide antibiotic for a medical condition (this can be a temporary exclusion if the drug is later discontinued) |
| ▶ Participant's parent/guardian has provided informed consent prior to initiation of any study procedures | |
| ▶ Participant's parent/guardian intends to reside in the study area until infant is at least 6 months old | |

cohort, participants are randomised 1:1:1:1 to one of four possible treatment assignments (figure 1). Infants born to mothers enrolled in the mother–infant cohort are randomised when the mother is enrolled, that is, before the infant is born. Infants born to multiple gestation pregnancies are randomised to the same treatment group. In the infant-only cohort, infants are randomized 1:1 to receive azithromycin or placebo.

Participants are randomised at enrolment using barcoded labels that are preprinted with a sequential participant identification number and randomisation assignment. The randomisation assignment corresponds to a masked treatment bin, denoted by a letter of the alphabet, and preprinted on every study medication bottle. Participants, field workers and study investigators are unaware of the link between the bin letters and the treatment assignments.

### Intervention

Participants receive either azithromycin (Zithromax) or placebo, both donated by the Pfizer Corporation. Both have the same appearance and colour, but the placebo contains no active substance. Study medication is reconstituted at the time of dosing and administered under direct observation in oral suspension form.

Pregnant participants are dosed with 50 mL of reconstituted liquid, equivalent to 2 g of study medication. Infants are dosed 20 mg/kg up to a maximum of 1 g, consistent with the recommended dosage for MDA.[10] Infant doses are calculated by the electronic data collection form according to the infant's weight at the study visit. Participants are observed for 30 min after receiving each dose. Participants who have an allergic reaction to the study medication are given medical attention as appropriate and are retained in the study but do not receive additional doses.

Dosing time points are aligned with routine healthcare visits. A minimum 72-hour window is required between doses. Pregnant participants receive up to three single doses over the course of the study, composed of 1–2 antenatal doses and a dose at delivery (table 2).

Ideally, pregnant participants are dosed in the second and third trimesters, but to accommodate individual variations in ANC timing, the first dose is administered at the enrolment ANC visit and the second dose at the next routine ANC visit or at delivery, whichever occurs sooner.

**Table 2** Study schedule

| Time point | ANC visit ≥14 weeks gestation | Second ANC visit | Delivery | 1 week after delivery | Penta-1 visit (approx. 6 weeks after delivery) | Penta-3 visit (approx. 14 weeks after delivery) | 6 months after delivery | 12 months after delivery* |
|---|---|---|---|---|---|---|---|---|
| **Mother–infant cohort participants** | | | | | | | | |
| Eligibility assessment and informed consent | M | | | | | | | |
| Randomisation | M/I | | | | | | | |
| Administration of study medication | M | M | M | | I | I | | |
| Demographic Questionnaire | M | | | | | | | |
| Medical history | M | M | M | | M/I | I | | |
| Labour and Delivery Questionnaire | | | M/I | | | | | |
| Weight measurement | | | I | | I | I | | |
| Vital status assessment | | M | M/I | I | M/I | I | I | I |
| **Infant-only cohort participants** | | | | | | | | |
| Eligibility assessment, informed consent and randomisation | | | | | I | | | |
| Administration of study medication | | | | | I | I | | |
| Demographic Questionnaire | | | | | I | | | |
| Medical history | | | | | I | I | | |
| Weight measurement | | | | | I | I | | |
| Vital status | | | | | I | I | I | I |

*Infants who reach 12 months of age before the end of follow-up have a final vital status assessment at 12 months. All other infants have a final vital status assessment at the end of the study follow-up period when they are between 6 and 12 months of age.
ANC, antenatal care; I, infant participant; M, mother participant; Penta-1, first pentavalent vaccination; Penta-3, third pentavalent vaccination.

The delivery dose is given during labour when possible but may be administered within 24 hours after delivery regardless of whether the infant is stillborn or liveborn. Infants are dosed when they present to a study facility to receive their first and third doses of pentavalent vaccine. Infants must be at least 6 weeks of age to receive the first dose of study medication.

### Follow-up and primary outcome measurements
The primary study outcomes are a composite of stillbirths and infant deaths for aim 1A, and infant deaths for aim 1B. Stillbirths are defined according to the WHO early stillbirth definition, that is, the birth of a fetus with no signs of life at ≥22 weeks gestation, or the birth of a fetus with no signs of life weighing at least 500 g.[18] Birth outcomes are recorded for every enrolled participant at the delivery visit. Participants who miss the delivery visit are followed up at least three times to ascertain the birth outcome.

Infant deaths are defined as any death of a liveborn infant in the first year of life, with infant vital status assessed at multiple time points (table 2). All enrolled infants are followed through at least 6 months of age. Study follow-up ends when the last enrolled infant reaches 6 months of age. At that point, all infants between 6 and 12 months of age will have a final vital status assessment. Infants who reach 12 months of age before the end of the trial will have a final vital status assessment at 12 months. Participants with a 6-month vital status that is unknown after at least three contact attempts by the study team are considered lost to follow-up.

### Secondary outcome measurements
Secondary outcomes include low birth weight (<2500 g), gestational age at birth (online supplemental appendix table 3), neonatal death (0 to <28 days) and a composite outcome of maternal hospitalisations and mortality through 6 weeks post partum. The cost-effectiveness of the interventions will also be estimated as a secondary aim.

### Safety monitoring and adverse events
Azithromycin is considered safe for use in pregnancy and is recommended for the treatment of urethritis and cervicitis caused by *Chlamydia trachomatis* and *Neisseria gonorrhoeae*.[19] Increased risks of malformations have been reported in some observational studies following first trimester exposure to azithromycin or to any macrolide antibiotic,[20–24] but not in others.[25–28] In a meta-analysis, azithromycin use (vs alternative antibiotics) in the first trimester was weakly associated with gastrointestinal malformations.[29] In the second and third trimesters, macrolide exposure was associated with genital malformations, comprised mostly of hypospadias.[20] One

case–control study reported increased relative odds of miscarriage in the first 20 weeks of pregnancy among those prescribed azithromycin.[30] A limitation of these observational studies is the potential for confounding by indication. In SANTE, participants are enrolled after the first 14 weeks of pregnancy and study personnel are trained to identify and report major congenital malformations as serious adverse events (SAEs). Other events considered as SAEs include pregnancy losses before 22 weeks, maternal deaths, hospitalisations and other significant or life-threatening illnesses. SAEs are reviewed by study physicians and assessed for relatedness to study medication.

In young children, the most common adverse reactions to azithromycin are gastrointestinal symptoms, reported in 1%–4% of those treated.[10 31] Cardiac toxicity is a potential concern, although the risk is not well established in paediatric populations.[10] A randomised trial of 450 children aged 6 weeks to 59 months found no severe adverse events in azithromycin-treated children, and only vomiting was reported more frequently in the azithromycin arm.[32] An association between treatment with erythromycin and infantile hypertrophic pyloric stenosis (IHPS) has been reported in the first 2 weeks of life, and to a lesser extent, in the third through fifth weeks of life.[33] The relationship between azithromycin and IHPS in infants less than 6 weeks of age is unclear.[34] In a recent RCT, one case of IHPS was reported among more than 10 000 azithromycin-treated infants fewer than 6 months of age.[35] The first dose of study medication in SANTE is administered at age 6 weeks or greater, although breastfed infants born to enrolled mothers could be exposed to low levels of azithromycin following the maternal delivery dose.[36] IHPS is monitored in SANTE as an adverse event of special interest and symptoms are solicited at all infant visits. An algorithm prompts physician review and assessment for relatedness, with any instances of probable or confirmed IHPS reported as SAEs (online supplemental appendix figure 1).

To detect potential harmful associations between azithromycin and the SANTE study endpoints, an independent statistician analyses stillbirths and infant deaths by treatment group at prespecified intervals. The integrity of the trial is maintained by limiting these analyses to the harmful side of the sampling distribution of the hazard and risk ratio statistics. Results are reported to the data safety and monitoring board (DSMB).

## Statistical considerations
### Sample size and power
The original power calculations for the primary aims were based on anticipated stillbirth and infant mortality rates in the Sikasso/Koutiala regions estimated from published data at the time the trial was being designed (table 3). Analysable endpoints were expected for at least 85% of enrolled participants. After the trial was initiated, observed stillbirth and infant mortality rates in the study population, pooled across blinded treatment arms, were

**Table 3** Sample size assumptions and minimum detectable relative reductions for the coprimary aims

|  | Original | Revised |
|---|---|---|
| Target sample size, mother–infant cohort | 33 600* | 49 600 |
| **Aim 1A. Effect of intervention on composite outcome of stillbirths and infant deaths from birth to 12 months** | | |
| Baseline combined stillbirth and infant mortality rate | 105/1000† | 74/1000† |
| Minimum detectable relative mortality reduction, comparing azithromycin to placebo treatment arms | | |
| 80% power | 9.5% | 8.7% |
| 90% power | 10.9% | 10.1% |
| **Aim 1B. Effect of intervention on infant deaths from first dose (approx. 6 weeks) to 12 months** | | |
| Infant mortality rate | 28/1000‡ | 9/1000‡ |
| Minimum detectable relative mortality reduction, comparing azithromycin to placebo treatment arms | | |
| 80% power | 18.6% | 24.7% |
| 90% power | 21.4% | 28.3% |
| With addition of infants from the infant-only cohort§ | | |
| 80% power | 16.6% | 22.3% |
| 90% power | 19.1% | 25.6% |

*In the original sample size calculation, 15% anticipated loss to follow-up (LTFU) meant that 28 560 infants were expected to have analysable endpoints.
†Original assumed composite baseline stillbirth and infant mortality was 105 deaths per 1000 births, calculated from 33 stillbirths/1000 births[55] and 74.8 infant deaths per 1000 live births (Sikasso data, Mali 2012 Demographic and Health Survey [DHS] report[56]), assuming that 6–11 months mortality data are available for 90% of infants and subtracting stillbirths from the live birth denominator. Revised observed composite stillbirth and infant mortality rate in the study population is 74 deaths per 1000 births.
‡Original baseline infant mortality rate after 6 weeks of age was 28 per 1000 live births (Sikasso data, 2012 Mali DHS report) assuming that 6–11 months mortality data are available for 90% of infants; revised infant mortality rate after 6 weeks of age was updated to 9 deaths per 1000 live births based on observed mortality in the study population.
§Original target sample size for infant-only cohort was 8500, assuming 7650 analysable (10% LTFU); revised target sample size 12 000.

used to update the sample size targets from 33 600 to 49 600 in the mother–infant cohort and from 8500 to 12 000 in the infant-only cohort (online supplemental appendix table 4). The resulting sample sizes are large, as infant mortality even in Mali is relatively rare, resulting in somewhat small absolute differences. For example, for Aim1B, with a baseline IMR of 9/1000, a 24.7% relative reduction means detecting an absolute reduction of less than 3 deaths per 1000.

### Analysis of primary aims
The primary analysis for aim 1A will be according to the intention-to-treat (ITT) principle. All live born infants

and all stillborn infants born at ≥22 weeks gestation or weighing at least 500 g will be included, with infants classified according to the treatment assignment made at the time of the mother's enrolment. For aim 1B, the primary analysis will be according to modified ITT, beginning from the first assigned dose. Sensitivity analyses of the two aims will be modified as-treated. If a mother or infant receives at least one study dose of azithromycin, they will be classified as being in the corresponding azithromycin-receiving study arm. Percent lost to follow-up will be compared across treatment arms for each intervention. Among those lost to follow up, baseline characteristics will be compared to assess whether loss is differential by intervention or by treatment arm. In the 2×2 factorial design, the primary analyses will fit a Cox proportional hazards regression model with two indicator variables for the two interventions, with censoring at the earliest of documented refusal for further participation, death, last study contact or recorded source of vital status, or attainment of 12 months of age. The study is not powered to detect a statistical interaction effect between the two interventions, but if it is important ($p<0.05$ or ratio of HRs >1. 5 or <0.67), results for all four study arms will also be reported. For primary aim 1A, the analysis timeline will be calculated from delivery, with stillbirths assigned day 0.1. For primary aim 1B, the timeline will start at the receipt of the first infant dose. For a measure of overall public health impact on infant mortality, a supplementary analysis will be done with the timeline starting at the live birth.

### Interim analysis

There is one prespecified formal interim analysis for efficacy planned for the maternal intervention, to take place when half of the expected number of stillbirths and infant deaths have occurred. This analysis will be run with an interim type I error of 0.007, which is expected to require a maternal intervention relative efficacy of at least 13%. Due to the complexity of this 2×2 factorial design, if this level was reached, the DSMB would consider all available data, and decide whether to stop randomisation to one or both interventions.

### Cost-effectiveness analysis

We will conduct a cost-effectiveness analysis of maternal, infant and combination azithromycin delivery, reflecting the arms of the trial. The economic cost of programme implementation will be estimated by summing the market price of azithromycin with costs of drug storage and personnel time for administration. Health outcomes will include maternal and infant mortality as measured for each arm, converted into disability-adjusted life-years based on life tables for Mali. We will calculate incremental cost-effectiveness ratios compared with the status quo and for progressively expensive strategies.

### Ancillary studies

Ancillary studies will be carried out in subsets of participants. Longitudinal sampling of mother–infant pairs will measure antimicrobial resistance of *Streptococcus pneumoniae* and *Escherichia coli* isolates and identify genomic markers of resistance. The impact of coadministration of azithromycin with rotavirus vaccine will be assessed. Several ancillary studies are aimed at elucidating the mechanisms by which azithromycin might prevent stillbirths and infant deaths, including investigations of the impact of azithromycin on maternal and infant anthropometry, the vaginal microbiome, the maternal and infant gut microbiomes, biomarkers of environmental enteric dysfunction, and malaria in pregnancy and infancy.

### Data collection and management

All study data are collected electronically on tablets using the open-source Tangerine data collection platform. The data collection platform is developed and maintained by RTI International, who serves as the data coordinating centre (DCC). Data are collected offline, and tablets are regularly synced to a cloud-based server for central aggregation and processing. Data are managed collaboratively by the DCC and study teams.

### Community engagement and consent

Community engagement was initiated in the study planning phase and will continue throughout the trial period. Engagement takes place through a series of meetings including local health centre personnel, religious and cultural leaders, and other community stakeholders at the regional and district levels. At each meeting, the investigators explain the trial procedures, the risks and benefits of participation, and other relevant information. A dynamic exchange ensues, and permission from the community leaders must be obtained to initiate and continue the trial.

### Ethics and dissemination

Written informed consent is obtained from the participant or their guardian. The study is approved by the ethics committees representing the University of Maryland, Baltimore and the Faculté de Médecine et d'Odonto-Stomatologie in Mali. Results of the trial will be shared with the communities where participants were recruited, Mali's Ministry of Health, the WHO and other stakeholders. A data sharing policy will be implemented. Results will be published in the peer-reviewed literature.

### Study oversight and quality assurance

A DSMB composed of experts in paediatrics, obstetrics, biostatistics and research ethics oversees patient safety and data collection for the trial. External monitoring visits are conducted quarterly. The study medication is analysed on a routine basis by an independent laboratory.

### Patient and public involvement

Local stakeholders in Mali will be involved in the research dissemination plans.

## DISCUSSION

The SANTE trial is poised to address important knowledge gaps regarding the potential for azithromycin to improve pregnancy outcomes and infant survival in a high mortality setting. Unique attributes of SANTE compared with other RCTs of azithromycin in pregnancy include (1) a large sample size with statistical power to detect a reduction in stillbirths and infant mortality, (2) a combined antenatal and intrapartum intervention, and (3) the assessment of maternal azithromycin as a stand-alone randomised intervention rather than one given in combination with other antibiotics or as part of an iPTP regimen.

With antibacterial, antiprotozoal, antimalarial, immunomodulatory and possibly antiviral properties, azithromycin delivered antenatally could prevent stillbirths and neonatal deaths by acting on causal or contributing factors including maternal infection, sterile or infection-related inflammation and poor maternal nutritional status.[37–41] Support for these hypothesised mechanisms of action is provided by randomised trials showing a reduction in low birth weight and preterm birth following antenatal azithromycin.[42–44] In The Gambia, a single 2 g dose of intrapartum azithromycin reduced bacterial carriage and the incidence of non-severe infection in neonates.[15 45] Whether this finding translates to reduced neonatal sepsis and death is being assessed in a follow-up trial (NCT03199547). A separate multisite trial will assess whether 2 g intrapartum azithromycin prevents sepsis and death in neonates and their mothers (NCT03871491). We did not identify any published RCTs of azithromycin in pregnancy with stillbirth or infant death as the primary aim. Those that included a stillbirth outcome as a non-primary aim found no effect.[44 46 47] Three trials reported a reduction in neonatal[44 48] or postneonatal mortality[49] while others found no difference by mortality by treatment arm.[15 42 46 50 51]

Evidence of azithromycin's effect on child mortality comes from three large, randomised trials. The first reported a 49% reduction in all-cause mortality (95% CI 10% to 71%) among Ethiopian children 1–9 years residing in communities randomised to receive MDA-azithromycin compared with those in control communities.[11] The second, known as the MORDOR trial, found a 13.5% (95% CI 6.7% to 19.8%) reduction in mortality in children 1–59 months and a 24.9% (95% CI 10.6% to 37.0%) reduction in infants 1–5 months who resided in communities randomised to receive biannual MDA-azithromycin compared with placebo.[12] Although MORDOR was designed to detect a mortality effect across sites in three countries, a statistically significant benefit was only observed in Dosso, Niger, the site with the highest under-5 mortality. To maximise the potential benefit of the infant intervention in SANTE, we situated the study in a region of Mali where the infant mortality rate is similar to that of the Dosso region.[52 53] Targeted administration earlier in infancy, when the risk of death is greater, and the addition of a second dose within the first 6 months of life could potentially increase the survival benefit for infants in SANTE above that observed in MORDOR.

In contrast to the MDA-azithromycin trials, Chandramohan *et al* reported no reduction in hospitalisations or mortality among children aged 3–59 months in Burkina Faso and Mali who were randomised at the household level to receive monthly SMC in combination with either azithromycin or placebo during the malaria transmission season.[13] One potential explanation for the difference in findings is that mortality reduction in the MORDOR sites, where routine SMC was not implemented, might have been due to azithromycin's antimalarial properties, and that there was no additional benefit of azithromycin beyond concurrent SMC in the Chandramohan trial. Children 3–59 months residing in the SANTE study area will be eligible to receive SMC during the malaria transmission season, but there are several reasons that SMC might be less influential in the SANTE trial. First, most infant deaths occur before 3 months of age. Second, only a subset of infants enrolled in SANTE will reach 3 months of age during the malaria season and thus be eligible for SMC. Third, only 54% of caregivers in Sikasso reported that their age-eligible child received one or more SMC doses during the malaria season.[16] This suggests that SMC coverage among infants enrolled in SANTE will be considerably lower than in the Chandramohan trial, where it was administered to all participants as part of the study intervention.

Most recently, the NAITRE trial in Burkina Faso individually randomised infants at 8–27 days of age to receive a single oral dose of 20 mg/kg azithromycin or placebo and found no reduction in mortality through 6 months of age.[35] Important differences between SANTE and NAITRE include different dosing schedules, higher baseline infant mortality in the SANTE study area, and the inclusion of very remote populations in SANTE.

The SANTE trial has two important limitations. The first is that ascertaining a reliable cause of death is not possible in the rural study setting. This limits inferences that can be made about differences in cause of death by treatment arm, although SANTE will conduct ancillary studies to understand possible mechanisms by which azithromycin might prevent mortality. Second, SANTE's individually randomised design will likely not achieve the interruption of pathogen transmission that might have occurred in MDA trials. One reason why a significant survival benefit for azithromycin has so far only been demonstrated in trials that randomised at the community level could be that the mechanism of protection requires high intervention coverage to interrupt pathogen transmission.[54]

In summary, SANTE is a large 2×2 factorial individually randomised placebo-controlled trial in a high mortality setting designed to generate evidence regarding the use of prophylactic azithromycin in pregnancy and early infancy to prevent stillbirths and infant deaths.

**Author affiliations**
[1]Center for Vaccine Development and Global Health, University of Maryland School of Medicine, Baltimore, Maryland, USA
[2]Centre pour le Développement des Vaccins, Bamako, Mali
[3]Obstetrics and Gynecology, Columbia University School of Medicine, New York, New York, USA
[4]Johns Hopkins Bloomberg School of Public Health, Baltimore, Maryland, USA

**Acknowledgements** The authors would like to acknowledge the following individuals for their many contributions to this work: Bill & Melinda Gates Foundation—Laura Lamberti. Data and Safety Monitoring Board—Baylor College of Medicine—Flor Muñoz-Rivas; University of Washington School of Medicine - Linda Eckert; University of Mali, Bamako—Allassane Dicko; Faculté de Médicine Pharmacie et d'OdontoStomatologie—Saïbou Maiga; University of California at San Francisco—Travis Porco. SANTE Study Group—University of Maryland, Baltimore—Wilbur Chen, Fleesie Hubbard, Yuanyuan Liang, Jacques Ravel, Ousmane S. Sow, Sharon Tennant, Mark Travassos; Centre pour le Développement des Vaccins, Bamako, Mali–Mamadou Diallo, Jane Juma, Awa Traore—RTI International - Matthew Finholt-Daniel, Emily Hadley, Jennifer J. Hemingway-Foday, Chris Kelley, Alice Litavecz, Elizabeth McClure, Melissa Page, David Plotner, RJ (Steinert) Corwin, Kevin Wilson; PATH–Robert Choy; Institute for Health Metrics and Evaluation, University of Washington, Seattle, Washington, USA–Michael Arndt.

**Contributors** KK is the principal investigator of the trial. KK conceived and designed the trial with substantial contributions from SOS, AD and LHM, in addition to FCH, ELD and MDT. AD, FCH, MDT, ELD, MF, RLG, LHM, SOS and KK wrote the protocol. LHM wrote the statistical analysis plan. AJD, FCH, MDT, ELD, OSS, TB, JAB, MK, DN, AMS, RLG, UO, SOS and KK developed data collection materials. SOS and FCH will lead the implementation of the trial with contributions from AD, KK, MDT, OSS, TB, MK and UO. AD drafted and revised the manuscript, with input from KK. All authors critically reviewed and approved the final manuscript.

**Funding** This work was supported by the Bill & Melinda Gates Foundation grant number 1196125.

**Disclaimer** The funders played no role in the design, writing, or decision to publish the study protocol.

**Competing interests** LHM serves on the Data Safety and Monitoring Board for Pfizer COVID-19 and respiratory syncytial virus vaccine trials.

**Patient and public involvement** Patients and/or the public were involved in the design, or conduct, or reporting, or dissemination plans of this research. Refer to the Methods section for further details.

**Patient consent for publication** Not applicable.

**Provenance and peer review** Not commissioned; externally peer reviewed.

**ORCID iD**
Amanda J Driscoll http://orcid.org/0000-0003-1385-960X

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
