## [Reviewer comments · BMJ Open]

ARTICLE DETAILS

TITLE (PROVISIONAL)	Antenatal, intrapartum, and infant azithromycin to prevent stillbirths and infant deaths: study protocol for SANTE, a 2x2 factorial randomized controlled trial in Mali
AUTHORS	Driscoll, Amanda; Haidara, Fadima; Tapia, Milagritos; Deichsel, Emily L.; Samake, Ousmane S.; Bocoum, Tiecoura; Bailey, Jason A.; Fitzpatrick, Meagan; Goldenberg, Robert; Kodio, Mamoudou; Moulton, Lawrence H.; Nasrin, Dilruba; Onwuchekwa, Uma; Shaffer, Allison M.; Sow, Samba O.; Kotloff, Karen

VERSION 1 – REVIEW

REVIEWER	Abdul Razak Princess Noura Bint AbdulRahman University
REVIEW RETURNED	13-Sep-2022

GENERAL COMMENTS	Thanks for the opportunity to review this RCT protocol. Overall, the protocol is well written - here are my few comments. Strengthen the introduction: Why azithromycin given to mothers should reduce stillbirths? What is the rate of stillbirths contributed by infection? Which stillbirth infections are susceptible to azithromycin and what is the rate of this infection? Same thing from neonatal perspective.. Methods Will mothers who have positive GBS swab will be included? How will you choose 8500 babies out of 33600 women? What about twin or multiple births? Will both babies (twins) be included? If yes, whether both babies will be randomised to same group or different group? What will happen to babies if they deliver premature? Will they receive the neonatal intervention? How will you handle mothers not receiving second antepartum dose if they deliver babies prematurely? How will you handle babies not receiving infant dose but have died in first 6 weeks? Sample size details are not clear - there is just a table and I could not understand how is this performed. For example what is the baseline rate of primary outcome? How much absolute difference in
---

	primary outcome are have authors powered? How was this difference considered meaningful? Were consumers or patients involved in making these decisions? Same thing for the infant death sample size. Please provide details of interim analysis. What are the safety endpoints? When will you consider to stop the trial prematurely? How was this identified? Analysis: How will the missing data be handled? Please provide more details on randomization and blinding in the appendix. How was the randomization generated? Was this stratified? Intervention: What are the differences in odor and taste between the placebo and the medication formulation? The trial is powered for efficacy end points but what about powering it to safety end points? Please provide details of protocol amendments until now.
--	--

REVIEWER	Janet Berrington Newcastle Upon Tyne Hospitals NHS Foundation Trust, Neonatology
REVIEW RETURNED	24-Oct-2022

GENERAL COMMENTS	This protocol paper is well written and balanced. It describes an ongoing large important study addressing optimal use of azithromycin in pregnancy and infancy. From the registration site the trial started in Sept 2020. I cannot see dates of the study in the manuscript, which BMJ Open state should be there. The abstract would also benefit from stating the setting that azithromycin has shown benefit (Introduction).
--

VERSION 1 – AUTHOR RESPONSE

Reviewer: 1

Dr. Abdul Razak, Princess Noura Bint AbdulRahman University

Comments to the Author:

Thanks for the opportunity to review this RCT protocol. Overall, the protocol is well written - here are my few comments.

Comment 1.1. Strengthen the introduction: Why azithromycin given to mothers should reduce stillbirths? What is the rate of stillbirths contributed by infection? Which stillbirth infections are susceptible to azithromycin and what is the rate of this infection? Same thing from neonatal perspective.

Response 1.1.

We thank the reviewer for his close review of our manuscript and for providing suggestions for improvement. We have added information to the introduction about the proportion of neonatal deaths and stillbirths that are estimated to be caused by infection and could thereby be potentially averted by prophylactic treatment with azithromycin. The proportion of stillbirths in LMICs that are caused by infection is not well described, but we added reference to a study that estimated that 17% of stillbirths in LMICs could have a pathogen in the causal chain that may be susceptible to azithromycin. In addition, there could be indirect (immunomodulatory) effects. In the discussion, we describe potential mechanisms whereby azithromycin could prevent stillbirths and infant deaths. We plan to delve further into this topic when we publish our trial results.

Comment 1.2 Will mothers who have positive GBS swab will be included?

Response 1.2 Yes, all women who meet the gestational age eligibility requirement, who plan to reside in the study area at least 6-months after their baby is born, who are not in active labor, and are not currently being treated with azithromycin will be included. Group B streptococcus screening is not standard of care for pregnant women in Mali, and we will not routinely test for this as part of study procedures. However, a subset of women (n=500) will have antenatal, intrapartum and postpartum vaginal swabs collected, and GBS results from these swabs will be analyzed as part of our comparison of the vaginal microbiome among women who are randomized to azithromycin treatment compared to those randomized to placebo treatment.

Comment 1.3 How will you choose 8500 babies out of 33600 women?

Response 1.3 We thank the reviewer for this question. The infants of all 49,600 women (sample size updated in revised manuscript) enrolled in the mother-infant cohort will be followed for 6-12 months. Separately, we will enroll 12,000 (revised from 8,500) additional infants, recruited at the vaccination visit when the first dose of pentavalent vaccine is received. The purpose of this additional infant cohort is to augment the sample size for primary aim 1B. To clarify this point, we have specified the sample sizes for both the mother-infant cohort and the infant-only cohort in the *Study Design, Objectives and Hypotheses* section of the Methods.

Comment 1.4 What about twin or multiple births? Will both babies (twins) be included? If yes, whether both babies will be randomised to same group or different group?

Response 1.4 We thank the reviewing for raising this question. Multiple births are included and randomized to the same treatment group. We have added the following text to the methods: "Infants born to multiple gestation pregnancies are randomized to the same treatment group".

Comment 1.5. What will happen to babies if they deliver premature? Will they receive the neonatal intervention?

Response 1.5 Enrolled women will receive the intrapartum dose (within 24 hours of delivery) regardless of whether the fetus is live born and regardless of gestational age. All infants receive the first dose of study medication at the time they receive the first pentavalent vaccination and must be at least six weeks of age to receive the dose. The dose is given regardless of gestational age at birth. To clarify this, we have added the following language to the “intervention” section of the methods: *Infants must be at least six weeks of age to receive the first dose of study medication.*

Comment 1.6. How will you handle mothers not receiving second antepartum dose if they deliver babies prematurely?

Response 1.6 We thank the reviewer for this question. Because study visits are aligned with routine health care visits, there are multiple scenarios that could result in an enrolled mother receiving only one antepartum dose. As mentioned by the reviewer, an enrolled mother could deliver her baby preterm, prior to the next expected follow up visit. Another scenario could be enrollment late in the pregnancy when no additional follow up visits are expected. In the statistical analysis section, we have added the following clarifying language: *All live and stillborn infants born at ≥ 22 weeks gestation, or weighing at least 500g, will be included, with infants classified according to the treatment assignment made at the time of the mother’s enrollment.*

Comment 1.7. How will you handle babies not receiving infant dose but have died in first 6 weeks?

Response 1.7 For primary aim 1A, all infants will be included, as noted above. For primary aim 1B, the primary analysis will be according to modified ITT, beginning from the first assigned dose. Because infants must be 6 weeks of age to receive the first dose, any infant who dies in the first 6 weeks will be included in the analysis of primary aim 1A but not in the analysis of primary aim 1B.

Comment 1.8 Sample size details are not clear - there is just a table and I could not understand how is this performed. For example, what is the baseline rate of primary outcome? How much absolute difference in primary outcome are have authors powered? How was this difference considered meaningful? Were consumers or patients involved in making these decisions? Same thing for the infant death sample size.

Response 1.8 We appreciate the reviewer’s feedback on the sample size table. We have reorganized the table to hopefully make clearer, and we have also included sample size updates that were made midway through the trial. The power calculations were made based on a relative reduction in mortality. We have included language in the table for clarity (“Minimum detectable relative effect”

changed to “minimum detectable relative mortality reduction, comparing azithromycin to placebo treatment arms,”) and given an example absolute difference calculation. For both aims, an approximate 20% relative reduction was considered the minimum we should be powered to detect, based on the findings from the MORDOR study, which showed a ~25% reduction in mortality in infants 1-5 months who received biannual azithromycin delivered by mass drug administration compared to placebo. Mali is a country with limited resources, and one of our Principal Investigators (SOS), who is a former Minister of Health of Mali, considered it necessary to have a sizable impact given the multiple-dosing nature of the interventions. Although external stakeholders were not involved in the power calculations for this trial, they will certainly be included when programmatic decisions are made, taking into consideration the results from trials testing different azithromycin interventions across different settings.

Comment 1.9 Please provide details of interim analysis. What are the safety endpoints? When will you consider to stop the trial prematurely? How was this identified?

Response 1.9 We thank the reviewer for the question and have added information about the formal interim analysis for efficacy of the maternal arm (Statistical considerations, interim analysis). There is no pre-specified futility analysis, nor is there an interim analysis of safety endpoints. There are so many different possible data configurations in this 2x2 factorial trial, that we thought it best to regard the meeting of the interim analysis criterion as a trigger to have the DSMB decide among numerous possible recommendations. The DSMB reviews all safety data on a bi-annual basis. The DSMB also reviews potential associations in a harmful direction between azithromycin at the stillbirth and infant death endpoints at pre-specified intervals, as described in the safety section of the methods.

Comment 1.10 Analysis: How will te missing data be handled?

Response 1.10 We appreciate this question and have added information to the statistical considerations section to describe how participants lost-to-follow-up will be handled. As only supplementary analyses will be adjusted for covariates, we have not discussed handling missingness of covariates in this manuscript, which will be heavily dependent on observed missing data patterns.

Comment 1.11 Please provide more details on randomization and blinding in the appendix. How was the randomization generated? Was this stratified?

Response 1.11 We have added detail randomization and blinding details to the appendix.

Comment 1.12 Intervention: What are the differences in odor and taste between the placebo and the medication formulation?

Response 1.12 As mentioned in the methods, active drug and placebo are identical in appearance. We have not assessed odor and taste differences but were informed by Pfizer that active drug and

placebo formulations contain the same artificial flavors.

Comment 1.13 The trial is powered for efficacy end points but what about powering it to safety end points?

Response 1.3 We appreciate the reviewer's question. The trial is not powered for safety endpoints. However, the large sample size will allow for the comparison of common SAEs across the treatment arms, most of which are expected to be more common than the primary efficacy outcomes. Infantile hypertrophic pyloric stenosis (IHPS) is a rare event which we will likely not have statistical power to assess across treatment arms; however, we have made a specific effort to capture IHPS cases when they do occur and expect that the SANTE trial will contribute valuable information regarding the risk of IHPS in neonates following exposure to azithromycin through breastmilk or the placenta.

Comment 1.14 Please provide details of protocol amendments until now.

Response 1.14 We have added a summary of major protocol changes as a table in the supplemental appendix

Reviewer: 2

Dr. Janet Berrington, Newcastle Upon Tyne Hospitals NHS Foundation Trust, Newcastle University
Faculty of Medical Sciences

Comments to the Author:

Comment 21. This protocol paper is well written and balanced. It describes an ongoing large important study addressing optimal use of azithromycin in pregnancy and infancy.

From the registration site the trial started in Sept 2020. I cannot see dates of the study in the manuscript, which BMJ Open state should be there.

The abstract would also benefit from stating the setting that azithromycin has shown benefit (Introduction).

Response 2.1 We thank the reviewer for her review and helpful feedback. We have added study dates to the study design, objectives, and hypotheses section of the methods, and have added information to the abstract about the study settings where the benefit of azithromycin has been shown prior to our trial.

Reviewer: 3

Prof. David Mabey, Department of Clinical Sciences, London School of Hygiene and Tropical
Medicine

Comments to the Author:

Comment 3.1 Given that the incidence of stillbirth is a primary outcome of this study and the fact that a study in Tanzania showed that 50% of stillbirths were due to untreated syphilis*, it would be helpful to know if pregnant women recruited into this study will be tested for syphilis and treated appropriately if they test positive.

*Watson-Jones D, et al. Syphilis and pregnancy outcomes in Tanzania. 1. Impact of maternal syphilis on outcome of pregnancy in Mwanza Region, Tanzania. J. Infect. Dis. 2002; 186: 940-7.

Response 3.1

Response 3.1. We thank the reviewer for the review and for providing this helpful citation. No, syphilis testing is not a procedure nor a screening criterion in this study. Although data are sparse, available evidence from surveys of blood donors and other sources suggest that syphilis is very uncommon in Mali, as is HIV. Standard of care in Mali is for midwives to offer syphilis testing and make treatment decisions for pregnant women. Study staff do not provide routine prenatal care for participants. Nevertheless, recognizing the importance that syphilis may play in the interpretation of the results, the study team determines by chart review whether the participant was tested and treated for syphilis by her health center provider and records these findings onto the study data collection forms. Though the small subset of women who are enrolled in the microbiome substudy (n=500) may have syphilis, group B streptococcus, or other pathogens detected in a vaginal sample, these are research findings that are not validated for clinical use, and will not provide results in a timeframe that informs clinical decision making. We will communicate the overall results regarding the prevalence of sexually transmitted infections in our study population to the health authorities so they can take action if needed.